# Preparation, Characterization and Evaluation of Organogel-Based Lipstick Formulations: Application in Cosmetics

**DOI:** 10.3390/gels7030097

**Published:** 2021-07-19

**Authors:** Cloé L. Esposito, Plamen Kirilov

**Affiliations:** 1Faculté de Pharmacie, Université de Montréal, Montréal, QC H3T 1J4, Canada; cloe.esposito@umontreal.ca; 2Institut des Sciences Pharmaceutiques et Biologiques, Université de Lyon (UCBL), Biologie Tissulaire et Ingénierie Thérapeutique UMR 5305 et Vecteurs Colloïdaux et Transport Tissulaire, 8 Avenue Rockefeller, CEDEX 08, 69373 Lyon, France

**Keywords:** LMOGs, organogels, lipsticks, formulation, photoprotection, physicochemical analysis

## Abstract

1,3:2,4-Dibenzylidene-D-sorbitol (DBS) and 12-hydroxystearic acid (12-HSA) are well-known as low-molecular-weight organogelators (LMOGs) capable of gelling an organic liquid phase. Considering their unique chemical and physical properties, we assessed their potential effects in new lipstick formulations by discrimination testing; in vitro measurements of the sun protection factor (SPF); and thermal, mechanical and texture analyzes. DBS and 12-HSA were used to formulate four types of lipsticks: L1 (1% DBS), L2 (10% 12-HSA), L3 (1.5% DBS) and L4 (control, no LMOGs). The lipsticks were tested for sensory perception with an untrained panel of 16 consumers. LMOG formulations exhibited higher UVA protection factor (UVA-PF) and in vitro SPF, particularly in the 12-HSA-based lipstick. Regarding thermal properties, the 12-HSA-based lipstick and those without LMOGs were more heat-amenable compared to thermoresistant DBS-based lipsticks. The results also showed the viscoelastic and thermally reversible properties of LMOGs and their effect of increasing pay-off values. In general, the texture analysis indicated that 12-HSA-based lipstick was significantly harder to bend compared to control, while the other formulations became softer and easier to bend throughout the stability study. This work suggests the potential use of LMOGs as a structuring agent for lipsticks, paving the way towards more photoprotective and sustainable alternatives.

## 1. Introduction

Over the past few years, lipsticks have gained significant popularity as one of the most attractive small luxury items, and sometimes even used as an economic indicator, following “the lipstick effect” observed in economic recessions [1]. Enhancing one’s attractiveness and appearance through lipstick relates to a wide range of colors used to fulfill customers’ needs and expectations across generations and countries [2]. While makeup trends are constantly evolving, lipstick trends are shifting in response to an increase in consumers’ personal care demands for natural, sustainable and organic cosmetics [3]. Lipsticks are commonly composed of several different components, such as vegetable oils (castor oil, almond oil), mineral derivatives (Vaseline oil, white petrolatum), pigments and waxes, which are not only used for aesthetic purposes but can also act as bioactive agents in extreme weather, e.g., UV protection [4,5]. Consequently, acceptable lipstick formulations should meet the following criteria: (1) thermal stability with a melting point generally within 55–75 °C and humidity variation to extreme maxima [6]; (2) dermatologically safe; (3) pleasant smell and taste; (4) softening at lip temperature (32 °C); (5) sufficient mechanical and physical properties with strength to maintain its structural integrity [7]; and (6) improved appearance without any defaults (air bubbles, cracks, sweating occurring during preparation steps).

Waxes are widely used in cosmetic products due to their viscoelastic, thermal and mechanical properties offering a broad range of industrial applications (in the cosmetics, pharmaceutical and agri-foods sectors) [8]. Since a wide variety of waxes derived from animal or plants are affected by environmental concerns, customers are switching to vegan-friendly and natural renewable cosmetics. Therefore, vegetable-based low molecular organogelators (LMOGs) and natural waxes have become an important alternative strategy to replace trans and saturated fats as structuring agents of edible oils [9,10]. 12-Hydroxystearic acid (12-HSA) is derived from naturally occurring ricinoleic acid, a hydroxylated fatty acid, present at 90% in castor oil and listed as a renewable source [11]. 12-HSA can form a self-assembled fibrillar network (SAFiN) upon the interaction of fibrillar crystals via non-covalent bonds such as van der Waals interactions, π–π stacking bonds and hydrogen bonds, as well as produce firm gels at a low concentration of 1% *w*/*w* [12]. Given its high melting point of 76 °C, viscoelastic properties offering better spreadability and its ability to stabilize active molecules such as UVB blockers in sunscreen [13], 12-HSA has been used in cosmetic applications but is as-yet relatively unexplored for lipsticks. 1,3:2,4-1,3:2,4-Dibenzylidene-D-sorbitol (DBS) is a well-known organogelator that has been in use for over 100 years, which is employed in personal care products owing to its high melting point of 225 °C and can be a suitable thickener agent in the organic phase [14]. DBS esters have been developed for lipsticks; compared to waxes and pasty compounds, they exhibited less transparency, glossy effects, stability and strength, among others, than DBS derivatives [15]. These LMOGs system permit: (1) thermoreversibility and thermostability, offering many advantages in drug delivery and long-term shelf-life systems [16]; (2) particular mechanical strength and flexibility due to their viscoelastic properties [12]; and (3) low-cost and simple preparation allowing large-scale production [16]. Moreover, the viscosity of the lipsticks has to be tunable to a high enough degree to maintain the stability of their structural network, while possessing a shear-thinning effect under mechanical stress during applications onto the lips [17]. Various studies have characterized the rheological, mechanical, physical stability and thermal behaviour of these LMOGs from diverse formulations [14,18,19]. For instance, Toro-Vasquez et al. studied the influence of an organogelator derived from stearic and (R)-12-hydroxystearic acid on crystalline microstructure organization and its rheological properties at two cooling rates (1 and 20 °C/min) [19]. The authors noted that the gelator structure influenced the gel microstructural organization through weak intermolecular interactions which, on a large scale, could affect its rheological properties (viscoelasticity and thixotropy). Lai et al. investigated the viscoelastic properties, network microstructure and morphology of DBS-based organogels using scanning electron microscopy, polarized optical microscopy, and rheology [18]. The results showed that both temperature and organogelator concentration influenced the self-assembly of DBS organogel networks. Finally, the stability was related to the thermodynamic equilibrium and was higher with high storage modulus, G′. However, the investigation of different LMOGs’ physicochemical properties on the mechanical and thermal properties of lipsticks has not yet been performed. Descriptive sensory analysis is an extremely useful tool in sensory science for obtaining both the qualitative and quantitative perceptions of a group of panelists [20]. Although sensory analysis is well established in cosmetics as a single methodology to compare different formulations according to their sensory attributes, there are few articles combining texture analysis and sensory analysis [21,22]. The purpose of this study was to use LMOGs as a substitute structuring agent to replace part of the large amounts of waxes in lipsticks. LMOG-based formulations were compared to non-LMOG lipsticks, which instead contained waxes commonly used in lipsticks; these were characterized by descriptive sensory (spreadability, hardness, greasiness, opacity and glossiness) and textural analyses (hardness, stiffness, firmness and pay-off). Furthermore, the in vitro protection factor (SPF) as well as the rheological and thermal behavior of the different lipstick formulations were investigated.

## 2. Results and Discussion

### 2.1. Effect of Organogelators on In Vitro SPF and UVA-PF

In vitro SPF and UVA-PF were evaluated in all lipstick formulations. Since some organogels have “self-healing” or thixotropic properties, the formulations were able to easily spread and form a protective film layer over the skin rapidly when the pressure was withdrawn. In Figure 1A, the photoprotection efficiency of lipsticks is depicted through the UV-absorbing profile of L1, L2, L3, L4 and L5 formulations in the UVB (290–320 nm), UVA II (320–340 nm) and UVA I (340–400 nm). L2 formulation containing 10% 12-HSA as organogelator showed higher UV-absorbing ability in comparison with the other formulations containing DBS as organogelator (L1 and L3) or no organogelator (L4, L5). This difference in absorbance intensity could be related to the optical properties of the sticks, and in particular to their diffusing capacity. Indeed, organogels contained 10 wt% of 12-HSA, which made them very diffusing (transfused aspect) and increased their capacity to absorb the UV radiation [23].

In addition, within the absorbance range from 0.2 to 0.6, the UV-absorbing capacity of formulations containing DBS was greater than lipsticks without organogelator. Many studies emphasized that the level of protection provided by a formulation may not depend only on SPF but also on its absorption spectrum and UVA-PF [24]. Indeed, UVA is involved among others in the UV-induced immunosuppression, in the generation of DNA damage and in cellular signaling pathways that regulate responses to DNA damage in melanocytes [25]. Regulatory guidelines in the European Union recommend a minimum UVA-PF/SPF ratio of at least 1:3 and an in vitro critical wavelength (CW) ≥370 nm [26]. All formulations were in agreement with these regulations except for L4 and L5 formulations, which failed to provide a broad-spectrum filter. In vitro SPF and UVA-PF were calculated to be, respectively, in the range 290–400 nm (Figure 1B) and 320–400 nm (Figure 1C), indicating that formulations with LMOGs, and more particularly, the 12-HSA formulation, had the highest SPF compared with the others. On the other hand, SPF and UVA-PF of L3 were higher than L1 with an improvement of SPF value (15.0%) due to its higher DBS content, but this enhancement was statistically not significant. In light of these results, organogelators can offer photoprotective effects in the manner of UV filters based on their physical structure through non-covalent interactions. However, SPF-15 or higher is the recommended blocking strength to prevent risk of cancer or skin ageing [27]. To some extent, the combined use of natural broad-spectrum sunscreens should be considered for further investigations. Nevertheless, these findings provide additional evidence of the photoprotective effect of 12-HSA that appeared to increase the UVB radiation absorption of a sunscreen formulation of gelled particles containing an immobilized organic filter (2-ethylhexyl-p-dimethylaminobenzoate) due to the formulation diffusing capacity related to their viscous consistency [13].

### 2.2. Instrumental Texture Analysis (Hardness, Stiffness, Firmness and Pay-Off) and Stability Studies

Lipstick instrumental attributes (hardness, stiffness and firmness) obtained by texture profile analysis (TPA) were reported based on the measurement of the maximum force to compress the lipstick. The effect of lipstick ingredients, storage temperature and storage time during TPA were evaluated (Figure 2). Hardness depends generally on the type and amount of waxes in the formulation [21], the oil polarity [28] and the oil:wax ratio [29]. In all lipstick formulations, the oil:wax ratio was constant for L1 and L4 formulations but lower than L2 < L3 due to the addition of white petrolatum. Generally, hardness of L2 at 25 °C and L3 at 25/45 °C did not change significantly over 12 weeks. However, L2 at 45 °C, and L1 and L4 formulations at 25/45 °C became softer after 12 weeks (*p* < 0.05) (Figure 2A). Furthermore, the difference in hardness between L1 at 25 °C and 45 °C at weeks 4 and 8, L3 at 25 °C and 45 °C at weeks 4 and 12, as well as for L4 at 25 °C and 45 °C during the whole study was statistically significant and attributable to softer lipsticks at 45 °C (*p* < 0.05). L4 was statistically significantly weakened and easier to bend than the other formulations. A commercial formulation (L5) was also tested in a single time point on day 1 and at both temperatures, as a reference formulation. Hardness of L5 was similar to L3 at 25 °C (*p* > 0.05) and closest to L4 at 45 °C. Acceptable solid lipsticks for the consumers should achieve at least 30 g-force (gf) at 20 °C for 8.1 and 12.7 mm diameter stick, as reported for current patented lipsticks [30,31,32]. Consequently, hardness values of all lipsticks were considered acceptable since values were above those of the commercial formulation or above 30 gf at 25 °C. The material properties of DBS organogels (L1 and L3) and especially the formulation without organogelator (L4) were also affected by higher temperatures. Indeed, at higher temperatures waxes become softer and are responsible for weak network strength [33]. Interestingly, the presence of LMOG as well as higher organogelator concentration increased thermal stability of the global network. These results were previously confirmed in different studies reported by Esposito et al., wherein 12-HSA-based organogels heated up above Tgel temperature undergo a molecular reorganization due to disruption of physical interactions between organogelator molecules [16].

Stiffness of the sample refers to its ability to resist elastic deformation during the bending action and is dependent on the elastic modulus (Young’s modulus). At 25 °C and 45 °C, L4 was the less stiff lipstick throughout the study (Figure 2B). At 25 °C, the stiffness of L1 and L2 were generally higher compared to the others, meaning they were less flexible, whereas L3 values were closest to the control. At 45 °C, L4 became less stiff over the weeks, indicating that these lipsticks containing no DBS possess the greasier and lubricant properties of than those of wax/oils formulation [34]. Lastly, firmness of the lipsticks was also affected by the presence of LMOG molecules and especially their weak physical interactions with the oily phase in the gel network structure, as reported [35]. This mechanical property is an important parameter that can play a role in tuning pay-off, friction and softness [34]. L3 exhibited the highest firmness and values remained constant compared to all other lipsticks over 12 weeks at 25 °C (Figure 2C). L4 values at 25 and 45 °C were significantly lower than L1–L3 lipsticks (*p* < 0.0001), suggesting that L4 and even L5 (having the lowest firmness values at day 1) were the softest lipsticks during the study. This finding could be explained as the temperature increases near their melting point temperature. At 45 °C, L1 and L3 formulations exhibited significantly softer texture throughout the weeks (*p* < 0.05). As expected, the presence and higher concentration of LMOGs enhance firmness. However, as temperature increased, the firmness of LMOG-based formulations with low organogelator concentration decreased, probably due to oil losses from the network [36].

From a consumer’s point of view, a lipstick product must comply with desired mechanical strength and pay-off characteristics to withstand the quality standard and requirement for an easy-glide application. Regarding requirements, the lipstick should have at least a pay-off of 0.0001 gm/cm^2^ [37]. Although a lower pay-off results in low amount of product attributes (color, durability, coverage) transferred, a balanced low/high pay-off needs to be considered to reduce product consumption and undesired sensation (waxy buildup). In addition, photoprotective formulations have to fulfil various requirements such as a high pay-off and a firm consistency with respect to their efficient photoprotective capacity [38]. L2 > L3 > L1 showed the higher pay-off to fabric results, whereas L4 < L5 had lower amounts transferred on paper during the experiments (*p* < 0.0001) (Figure 3). However, it should be noted that all pay-off lipsticks had passed the minimum requirement. The higher pay-off values between LMOG-based and “classical” lipstick formulations could be explained by a higher frictional force change attributed to gel samples [39].

Considering the overall mechanical attributes, LMOGs have great potential to replace part of waxes in lipstick formulations and to contribute to attaining the desired textural properties for photoprotective cosmetics.

### 2.3. The Combined Use of Thermal and Rheological Analyses on the Dynamic Structure Network of Lipsticks

The temperature sweeps are displayed in Figure 4 and were performed in LVR regime to assess the effect of organogel-based lipsticks between 15 °C and 150 °C on the dynamic elastic behaviour (G′) and the dynamic viscous behaviour (G″). The organogel-based lipsticks exhibited thermoreversibility and viscoelastic properties. According to the classical viscoelasticity theory, the initial rise in G′ is due to 3D network formation, while onset sol-gel phase transition (Tgel) of each lipstick can be estimated from the abrupt increase in G′. Indeed, Tgel is determined by the G′/G″ crossover point from G′ < G″ to G′ > G″ of each cooling curve. A crossover point of G′ and G″ is identified as the gel melting point (Tm). Upon cooling, the resulting disordered fluid-like state of the gel undergoes a sol-gel transition generating a viscoelastic solid-like structure of crystalline organogelator molecules [12]. L1 and L3 showed a crossover point at high temperatures underlying a more stable network with highest melting stability [7,40], while L2 and especially L4 and L5 were more heat amenable and prone to soften at low temperature.

Tgel and Tmelt values determined in the rheological study were compared to Tm values obtained from melting point and are displayed in Table 1. The Tm values determined by melting point and rheology were in close agreement and were related to the breakage of particle interactions holding up the 3D structure. It was also noted a small but significant increase in the Tgel value (*p* value < 0.05) and in a maximal G′ of 12.9 times, with the addition of DBS organogelator at 1.5% (*w*/*w*) in the lipstick network (L3), compared with the control (L4) (Table 1). Higher G′ values are typically associated with stronger a three-dimensional gelled network [41].

The gelation point can also be observed with the tan δ (i.e., G″/G′, damping coefficient) at tan (δ) = 1, as a simple indicator. Appendix A show the loss tangent curves of the lipstick formulations (L1 to L5). At temperatures lower than 50 °C for L2, L4 and L5 or 129 °C for DBS-based lipsticks, respectively, tan (δ) < 1, indicating that the sample behaves in a manner more similar to a solid with elastic properties, while at tan (δ) > 1 the organogels are viscous liquid. When comparing gelation points obtained during the heating and cooling ramp test, there was a significant decrease (*p* < 0.05) in gelation points for L1, L2, L3, L4 and L5 formulations (14.5, 13.04, 3.47, 0.99 and 12.5%, respectively) for cooling compared to heating sweep. This difference between gelation point values during melting and crystallization processes highlights the hysteresis effect [41] and may be due to the heat dissolution of waxes and LMOGs clusters during the heating process [43], delayed kinetically by hydrogen-bond breaking. Nevertheless, all samples showed acceptable thermal stability and could be considered as heat-resistant lipsticks.

### 2.4. Relationships between Descriptive Sensory and Instrumental Analysis of Organogel-Based Lipsticks

All organogel-based formulations (L1–L3), as well as control (L4) and a commercial formulation (L5), have been evaluated for sensorial attributes as response with a panel of consumers. This quantitative description analysis (QDA) aimed to (i) characterize the sensory properties profile of the produced lipsticks and (ii) perform a principal component analysis (PCA) for mapping the main similarities and differences between samples and their sensory attributes. All tested attributes, spreadability, hardness, opacity, glossiness and greasiness, were tested by assessors through an objective evaluation and scored following the intensity of each attribute for each product (corresponding to a numerical value between 0 and 9). The performance of the panel was evaluated through different criteria: consensus (individual product evaluation) and discrimination between product and attributes. F-values and *p*-values, according to two-way ANOVA followed by Fishier LSD post hoc test, were appropriate to measure discrimination power and are reported in Table 2. Regarding panelist effect and attribute discrimination, panelists were able to significantly discriminate attributes reaching a consensus (*p*-values < 0.001), while all attributes were considered discriminating (*p*-values < 0.001) among the formulations. From mean ± standard deviation values (Table 2) and spider-plot of the evaluated attributes for each formulation (Figure 5A), greasiness and glossiness sensory attributes are the most discriminative ones, varying from 1.0 to 7.8 and 1.0 to 7.5, respectively. The L2 lipstick formulation showed a low glossiness effect compared to the other L1 and L3 DBS-based formulations. L3 lipstick containing the highest amount of vaseline oil (48.5% *w*/*w*) compared to other ingredients such as white petroleum (L1) and organogelator (L2) was assessed as the least greasy formulation. Medium desired greasiness was achieved for LMOG-based formulations (L1–L2) with an average of 5.6 points. All lipsticks expressed a good spreadability with non-significantly different values ranging from 6.7 to 8.1 points. The hardest and softest lipsticks were L2 and L1/L4, respectively, which was in agreement with texture analysis, except for L1 at day 1. This exception may have been caused by the lack of ingredients to provide a soft touch, despite a consensus reached between panelists with low value variability.

The PCA was also performed on both sensory and instrumentals of pick-up sensory properties (hardness, firmness, stiffness) data [44], to facilitate the selection of raw materials to formulate desired lipsticks (Figure 5B,C and Appendix A). The two principal components (PC1 and PC2) accounted for 93.1% of the total variance between samples. The plot used to read PCA scores of each sample (represented as dots) allows the identification of the positioning of the sensory attributes and the identification of the more relevant sensory descriptors close to + 1 and –1, and thus contributes mostly to PC (Figure 5B). As observed in Figure 5B, the glossy aspect (−0.969) contributed negatively to PC1, while hardness (0.750) contributed positively. Moreover, spreadability (0.791) contributed positively to PC2 whereas greasy effects (−0.859) contributed negatively. According to Buehler et al., correlation between two variables and two component analysis is shown as a vector, which indicates the strength of the relationships following the vector length and whether the correlation is positive or negative according to its direction [45]. Since the main attributes are identified, the loading plot (Figure 5C) displays more information about the relationships between sensory descriptors and lipstick formulations. It seems there is a covariance between different sensory attributes. Indeed, a higher hardness is generally associated with a lower glossy effect, as is observed for L2. It is also interesting to notice that all formulations are different from each other’s and do not form a cluster. L2 and L3 formulations are less related to greasy effect, while L3, L1 and L4 are related to glossiness.

Another loading plot (Appendix A) was performed combining both texture characteristics and sensory descriptors, with numerical variables represented as lines and dotted lines, respectively. All instrumental analysis—hardness (0.973), firmness (0.792), stiffness (0.907) and hardness form sensory analysis (0.657)—contributed positively to PC1, while opacity (−0.831) and the glossy effect (−0.630) contributed negatively (Appendix A). Furthermore, instrumental hardness is essentially correlated positively with firmness. According to the representation in Appendix A, formulations can be split into two different clusters of sensory and instrumental attributes: (1) L2 is characterized by instrumental measures, primarily hardness and firmness, while L4 is negatively correlated to these parameters; (2) L1 and L3 are much more related to firmness, and to some extent glossiness. Overall, the main discriminative attributes for the analyzed lipsticks were greasiness and glossiness. The experimental and sensory hardness parameters were in total agreement for the hardest and softest formulation.

## 3. Conclusions

The main objective of this study was to evaluate the effect of LMOGs, as a substitute alternative to replace part of waxes, on the thermal, rheological, sensory and performance (textural and photoprotection efficiency) properties of the lipstick formulations. Despite growing concerns about lip cancer and the importance of photoprotection, there is still a need for novel strategies to design LMOG lipsticks. Indeed, the in vitro SPF experiments showed that LMOGs and especially the 12-HSA lipstick formulation significantly increase in vitro SPF and UVA-PF compared to lipsticks based on waxes. The thermal properties of lipstick could be determined in agreement for the two methods employed (rheology, melting point) and showed that DBS-based organogels were more heat stable, which is directly correlated to the resistance of the organogel structure to deformation. Furthermore, all organogels displayed typical viscoelastic and hysteresis behaviour, that may be explained by the difference in the dynamic structure organization and disorganization through hydrogen bonding of the gel network.

Regarding texture characteristics, 12-HSA-based lipstick was significantly harder to bend than the control formulation (without organogelator) and stable at 25 °C, while other lipsticks (except L3) generally became softer for both temperatures throughout the long-term study (12 weeks). Finally, we found that the higher firmness was correlated to the presence and the increase in concentration of LMOGs at 25–45 °C. Any future investigations of the effect of LMOG lipsticks on friction during tribological studies will provide a potential correlation of glide (lubricating properties) with other sensory factors. By combining sensory attributes (glossy) and instrumental parameters (hardness, firmness), we were able to cluster the formulations. Considering the overall mechanical, thermal and photoprotection attributes, LMOG-based formulations have great potential to replace part of waxes in lipsticks, while offering tunable textural parameters to target a large number of customers.

## 4. Materials and Methods

### 4.1. Materials

12-HSA and DBS organogelators with 99% purity were obtained from Casid^®^ Vertellus (Greensboro, NC, USA) and Sigma-Aldrich (Saint-Louis, MO, USA), respectively. Vegetable and vaseline oils were bought from, respectively, BASF (Coptis, Lavallois-Perret, France), and Gattefossé (Saint-Priest, France). White petrolatum and beeswax were bought from Sigma-Aldrich. Nesatol and pigments were a gift from our industrial partners. Demineralized water was used for the lipstick preparations.

### 4.2. Preparation of Organogel Lipsticks

Lipsticks were formulated by the moulding method using a mixture of natural ingredients reported in Table 3. Firstly, LMOGs (Appendix A) and/or waxes were added into the oily phase, homogenized using a high-speed homogenizer) at a speed of 500 rpm [6]. The phase A was heated to either 125 °C or 200 °C when formulated with DBS. Phase B was composed of beeswax and pigments and heated at 100 °C to ensure the dispersion of pigments prediluted in Nesatol. Phase B components were added to phase A at the same temperature and homogenized together. Then, the mixture, still hot, was poured into a lipstick mold, kept at room temperature and away from direct sunlight. The resulting formulations are represented in Appendix A.

### 4.3. In Vitro Sun Protection Factor (SPF) Test Method

SPF was measured with a spectrophotometer (UV 2000 S, Labsphere, Bures sur Yvette, France) to analyze UV transmittance of samples. The lipsticks in the amount of 1.3 mg/cm^2^ were spread on polymethyl methacrylate plates (5 cm × 5 cm × 0.3 cm, Helioscience, Marseille) uniformly on their roughened surface, following the procedures present in ISO 24443:2012 [46]. Glycerin (15 μL) was used for the blank scan to calibrate the spectrophotometer before UV transmittance measurements. PMMA plates (*n* = 3 per sample) were then kept in a dark room for 15 min at 25 °C. Six different locations were measured for each sample and transmittance spectrum was measured in the range from UVB (290–320) to UVA (320–400 nm) at 1 nm intervals to determine the SPF value of each sample. In vitro evaluations of SPF and UVA-PF were calculated using the following equations [46]: (1)$$SPF{\;}_{in\;vitro}=\;\frac{\int_{290}^{400}E_{\left(\lambda \right)}\;I_{\left(\lambda \right)}\;d_{\left(\lambda \right)\;}}{\int_{290}^{400}E_{\left(\lambda \right)\;}I_{\left(\lambda \right)\;}10^{\;-\;A_{0}\left(\lambda \right)}\;d_{\left(\lambda \right)}}$$
where *E_λ_* is the erythema action spectrum at wavelength λ, *I_λ_* is the spectral irradiance of the UV source at wavelength *λ*, *A*_0_(*λ*) is the mean monochromatic absorbance measurements per plate of the test product layer before UV exposure at wavelength λ and *d_λ_* is the wavelength step (1 nm).

Each sample was exposed to the UV exposure dose, which is the initial UVA-PF (UVA-PF_0_) [46] value multiplied by a factor 1.2 (J/cm^2^):(2)$$UVA-PF=\frac{\int_{320}^{400}P_{\left(\lambda \right)}\;I_{\left(\lambda \right)}\;d_{\left(\lambda \right)}}{\int_{320}^{400}P_{\left(\lambda \right)\;}I_{\left(\lambda \right)\;}10^{\;-\;A_{e}\left(\lambda \right)C}\;d_{\left(\lambda \right)}}$$
where *P_λ_* is the persistent pigment darkening (PPD) action spectrum, *A_e_* is the mean monochromatic absorbance measurements plate of the test product after UV exposure and *C* is the coefficient adjustment determined using the SPF label (SPF in vitro adj [46] generated by the UV-2000’s software version 1.10).

### 4.4. Mechanical Properties

The mechanical properties of samples were evaluated by compression testing (bending and needle probe penetration tests) or tension–compression tests (cycle count test). Samples were kept at 25 °C and 45 °C for 24 h before testing. Visual inspection and measurements were taken at day 1, 2 weeks, 4 weeks, and 8 weeks using a texture analyzer (Ta.XT+; Texture Technologies Corp., Hamilton, MA/Stable Microsystems, Godalming, Surrey, UK). All samples had an inner diameter of 9 mm and a height of 35 mm. All measurements were carried out with six lipsticks of each type. The data (force, distance) were recorded using Exponent Stable Micro Systems software (version 8). The maximum force value, which provides the hardness of the sample, is the force required to bend the sample until it broke off at the base and for a defined distance, and this gives an indication of the brittleness of the sample. The stiffness of the sample was evaluated with the gradient of the slope during the bending action (Appendix A).

Bending test: Measurements to determine the bending force and stiffness were made using a hemispherical blade coming down at 3 mm away from the tip of the horizontally clamped sample (rolled out to its maximum length). The test and target type were set to “compression” mode and “distance”, respectively. All measurements were carried out at a trigger force of 20 g; pre-test, test and post-test speeds of 1.0, 1.0 and 10.0 mm s^−1^, respectively; and a target value of 10 mm.

Needle probe penetration tests: This method for testing lipstick hardness and firmness is adopted from the ASTM Standard Method of test D1321-10 using a 2 mm needle probe (TA39). Measurements were performed in the following conditions: (1) the test and target type were also set on “compression” mode and “distance”, respectively; (2) a penetration depth of 10 mm; (3) a trigger force of 5 g; (4) a pre-test, test and post-test speeds of 1.0, 2.0 and 10.0 mm s^−1^, respectively. Before the test, each lipstick was centered under the needle probe to facilitate its penetration.

Cycle count test: Pay-off test aims to determine the mass released while applying a lipstick. Pay-off is often used as an indirect measure of how well a product works from a consumer’s perspective. Briefly, a 1 cm portion of lipstick was removed, and the tip was rubbed on a piece of paper to obtain a homogeneous flat lipstick tip. The experimental setup consisted of a slot arrangement having a TTC vertical friction rig for holding the lipstick perpendicular to the paper strip (~7 cm wide) fixed at a stationary vertical plate to create a flat surface and separated with a 5 mm distance.

The cycle count test was performed for 3 cycles with a speed of 5 mm s^−1^ and a deformation distance of 60 mm. Each lipstick was pre-weighed and reweighed after the test, in order to calculate the pay-off value.

### 4.5. Thermal Properties

Melting point: The melting points of the lipsticks were determined by the drop-ball method (ASTM D127). Lipstick samples (2 g) were poured into a thin-walled capillary of hard glass, about 20 cm long, with an internal diameter of 0.6 mm and sealed at both ends. Each tube was dipped into a glass beaker containing water and a thermometer, and water was heated at a rate of 1 °C/min until the temperature reached 95 °C. A stainless steel ball with a diameter of 0.5 mm (1 g) was put on the surface of the lipstick. The transition temperature at which the material forms a liquid and the ball reaches the bottom is considered as the melting point (Tm).

### 4.6. Rheological Studies

For the rheological characterization of the lipsticks, a temperature ramp test was conducted in a stress-controlled rheometer (AR 1000, TA Instruments, New Castle, DE, USA) with parallel plate geometry (8 mm diameter and 0.7 mm gap). In order to prevent slippage of samples, both lower and upper plates were covered with sandpaper (the viscoelastic response of the sample was not affected by this layer). The rheometer was equipped with a Peltier plate regulating the temperature within ± 0.1 °C of the set value. The sample preparation steps were conducted as reported by Pan et al. [7]. Briefly, each uniform lipstick slice was placed between the two plates. To prevent dehydration and to maintain humidity of the sample, a thin layer of paraffin oil was placed at its surface and moist pieces of cotton were placed around each slice. The linear viscoelastic region (LVR) was determined thanks to an isothermal strain sweep (0.001% < γ < 100%) at a fixed angular frequency (ω = 1 rad s^−1^) to identify linear and non-linear regions. Within the LVR, the dynamic storage or elastic modulus (G′) and loss or viscous modulus (G″) are independent of the strain amplitude (1%). The temperature ramp test was performed in the LVR at the temperature range of 20–150 °C for all lipsticks, according to a melting point study at a heating rate of 1 °C min^−1^ and at a frequency of 1 Hz. This test aimed to evaluate the dynamic storage or elastic modulus (G′), loss or viscous modulus (G″) and the loss tangent tan *δ* during temperature variation.

### 4.7. Sensory Analysis

Sixteen female participants aged between 18 and 25 years who frequently used lipstick products were recruited for sensory evaluation. The overall sensory analysis was performed at room temperature (25 ± 2 °C/60% RH ± 5%). The samples were evaluated in randomized order and lipstick identified anonymously. Sensory analysis consisted of a nine-point hedonic scale (1–9) whereby sensory attributes spreadability, hardness, opacity, glossy and greasy effects were evaluated. Five designed lipsticks formulations (Table 3) were evaluated by the panel of female participants. A commercial lipstick (REF, L’Oréal Nude, TD023, Color Riche 235) was included to test the sensitivity of the recruited panels. Each panelist filled out questionnaires with parameters scored between 1 and 9, where 9 represented the highest intensity of the parameter.

### 4.8. Statistical Analysis

Analysis of variance was performed on the in vitro SPF, mechanical/rheological properties and pay-off using Graphpad Prism 7.0 and for sensory data using PanelCheck V1.4.2 software. The difference of SPF among lipstick formulations was analyzed for significance using one-way ANOVA and Tukey’s multiple comparison post hoc test. Regarding mechanical profiles and pay-off test, statistical tests were performed by paired *t*-test and/or one-way ANOVA followed by Tukey test. The difference of Tgel or Tm_rheology_ among lipstick formulations was analyzed for significance using the Kruskal–Wallis test and Dunn’s multiple comparison post hoc test. For sensory analysis, Friedman’s two-way ANOVA followed by Fisher’s LSD test was used with panelist and product effect interactions as variation factors. Principal component analysis (PCA) was applied to establish the relationship between attributes and lipsticks along with mean ratings of attributes from the sensory panel vs. those from the texturometer. A value of *p* < 0.05 was regarded as significant, and data were represented as average value ± SD.

## Figures and Tables

**Figure 1 gels-07-00097-f001:**
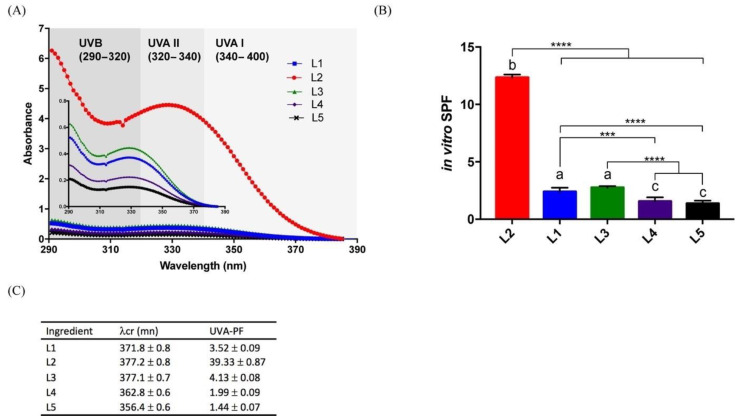
Ultraviolet photoprotective performance of lipsticks. UV absorbance spectra profile (**A**), in vitro SPF (**B**) and UVA protection factor (UVA–PF), critical wavelength (λcr) (**C**) SPF, λcr and UVA–PF values are expressed as mean ± SD of six experimental measurements per lipsticks. An asterisk indicates a statistical difference between formulations, ***: *p* < 0.001, ****: *p* < 0.0001, one-way ANOVA followed by Tukey’s multiple comparisons post hoc test for (**B**). Bars that do not share similar letters denote statistical significance, *p* < 0.05.

**Figure 2 gels-07-00097-f002:**
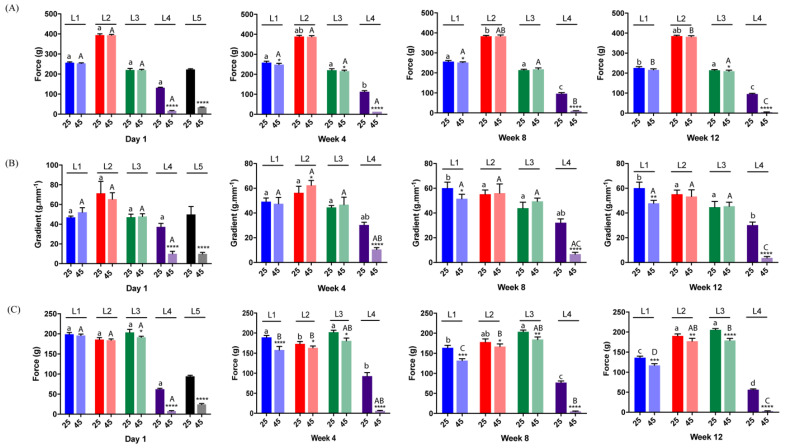
Mechanical profiles of lipsticks. Hardness (**A**), stiffness (**B**) and firmness (**C**) during stability studies carried out for 1, 28, 56, and 84 days, at 25 and 45 °C. Force values are expressed as mean ± SD of *n* = 6 lipsticks per group. Different lower-case letters indicate a significant difference for hardness, firmness or stiffness between days of the same formulation at 25 °C and different capital letters indicate a significant difference among different days of the same formulation at 45 °C (one-way ANOVA, *p* < 0.05). Asterisk indicates a statistical difference between 25 and 45 °C for the same formulation in the same day. *: *p* < 0.05, **: *p* < 0.01, ***: *p* < 0.001, ****: *p* < 0.0001, paired *t*-test.

**Figure 3 gels-07-00097-f003:**
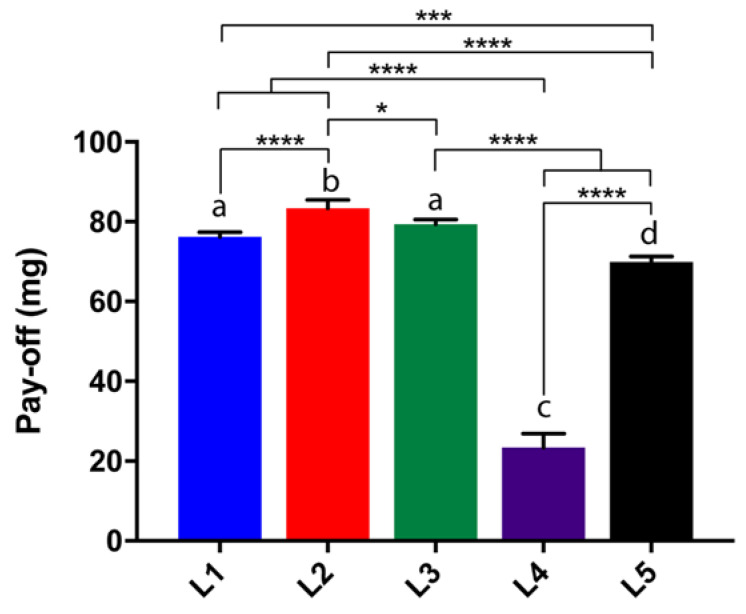
Pay-off to fabric of lipsticks (expressed in mg, mean ± SD, *n* = 6). An asterisk indicates a statistical difference between formulations. * *p* < 0.05, *** < 0.001, **** *p* < 0.0001, One-way ANOVA followed by Tukey’s multiple comparison test. Bars that do not share similar letters denote statistical significance, *p* < 0.05.

**Figure 4 gels-07-00097-f004:**
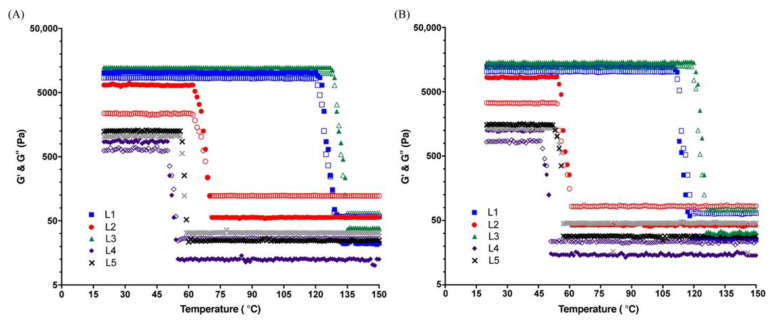
Temperature sweep experiments. Heating (**A**) and cooling (**B**) cycles for lipsticks L1 (blue squares), L2 (red circles), L3 (green triangles), L4 (purple diamonds) and L5 (black cross). Linear elastic (G′) and viscous (G″) moduli are shown as plain and open symbols, respectively. Each temperature ramp was performed with a heating rate of 1 °C min^−1^, at a fixed angular frequency of 1 rad s^−1^ and a 1% strain within the LVR regime.

**Figure 5 gels-07-00097-f005:**
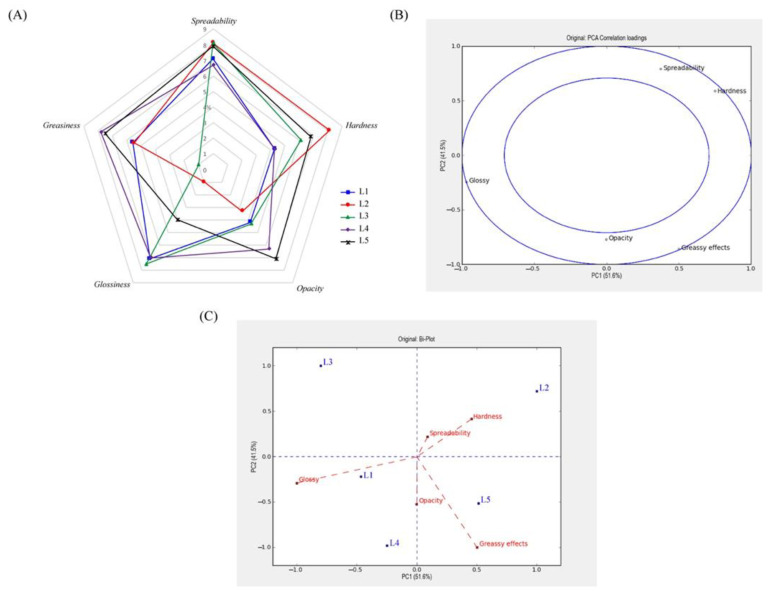
Sensory evaluation results of the produced and commercial lipsticks after application. Spider-plot (**A**), correlation loadings of the principal component analysis (PCA) for all the sensory attributes (**B**), and biplot representation of PCA illustrating relationship between attributes and formulations (**C**).

**Table 1 gels-07-00097-t001:** Gelling (Tgel), melting (Tm) temperatures obtained with rheology (determined as reported) [16,42], melting point technique and maximum G′ recordings of lipstick formulations.

Formulations	Tgel (°C)	Tm_rheology_ (°C)	Tm_melting point_ (°C)	Maximum G′ Value (Pa)	Temperature at Which Max. G′ Value Was Recorded (°C)
L1	119.6 ± 0.5	120.3 ± 0.6	>95	10,502	32
L2	61.8 ± 0.7	62.1 ± 0.4	62	6985	32
L3	127.2 ± 1.1	127.5 ± 0.5	>95	12,398	63
L4	50.7 ± 0.4	50.6 ± 0.7	49	963	37
L5	56.6 ± 0.4	55.7 ± 0.6	54	1295	36

**Table 2 gels-07-00097-t002:** Results of two-way ANOVA and Fisher’s LSD test examining the influence of product effects according to panel responses.

Formulations	Effects	Statistics	Attributes				
			Spreadability	Hardness	Opacity	Glossiness	Greasiness
L1		Means	7.1	4.3	4.2	7.1	5.6
		SD	0.64	0.92	1.43	0.87	1.64
L2		Means	8.1	8.1	3.3	1.0	5.5
		SD	0.82	0.82	1.01	0.0	1.77
L3		Means	8.1	6.1	4.3	7.5	1.0
		SD	0.82	1.33	0.92	1.09	0.0
L4		Means	6.7	4.3	6.3	7.0	7.8
		SD	1.01	1.24	1.13	1.09	1.24
L5		Means	7.9	6.8	7.1	4.0	7.5
		SD	0.96	1.15	0.64	1.53	1.37
	Panellist	F value	11.22	13.89	10.24	7.35	9.77
	Product		27.11	125.87	103.04	257.85	176.16
	Panellist	*p* value	<0.001	<0.001	<0.001	<0.001	<0.001
	Product		<0.001	<0.001	<0.001	<0.001	<0.001

**Table 3 gels-07-00097-t003:** The set of component formulations and functional categories of their ingredients.

Ingredients	Functional Category	Quantity (% *w*/*w*)				
		L1	L2	L3	L4	L5
**Phase A**						
Vaseline oil	Occlusive agent	0	40	48.5	0	L’Oréal Nude, TD023, Color Riche 235
Castor oil	Moisturizer	14	20	20	15	
Almond oil	Emollient	45	20	20	45	
White petrolatum	Lubricant	30	-	-	30	
DBS	LMOG	1	-	1.5	-	
12-HSA	LMOG	-	10	-	-	
**Phase B**						
Beeswax	Thickening agent	5	5	5	5	
Pigments	Colouring	5	5	5	5	

## Data Availability

The study did not report any data.

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
