# Peer review of "Preparation, Characterization and Evaluation of Organogel-Based Lipstick Formulations: Application in Cosmetics"

_gels, 2021, doi:10.3390/gels7030097_

Round 1
Reviewer 1 Report
This work describes the photoprotective properties of organogel-based lipstick formulations. While the goal of developing lipstick base alternatives with higher photoprotective properties is interesting, the work presented here lacks in scientific rigor.
- The choice of the two organogels (DBS and 12-HSA) is not justified. No premise is given as to why these two oranogels, at the studies concentrations, might be good candidates to increase photoprotection. The choice seems random.
- While the title and purpose affirm emphasis on photoprotection, the majority of the work is on mechanical properties
- Only 3 formulations were tested here, and the one that showed photoprotective properties was tested only at one concentration
- No discussion as to what makes a gelling material photoprotective is included
- The statement in the conclusions section "design photoprotective lipsticks" is inaccurate since no design was involved here.
More experiments and thoughtful experimental design are needed for this work to be meritorious.
Author Response
Manuscript ID Gels-1255378
Dear Reviewer,
Thank you for giving us the opportunity to submit a revised draft of the manuscript "Preliminary Study of the Potential Photoprotective Effect of Organogel-Based Lipstick Formulations: Texture Analysis, Rheological, Thermal and Sensory properties" for publication in Gels. We appreciate the time and effort that you and the reviewers dedicated to providing feedback on our manuscript and are grateful for the insightful comments on and valuable improvements to our paper. We have incorporated most of the suggestions made by the reviewers. Those changes are written in red within the manuscript. Please see below, in red, for a point-by-point response to the reviewers’ comments and concerns.
Response to Reviewer 1
- The choice of the two organogels (DBS and 12-HSA) is not justified. No premise is given as to why these two oranogels, at the studies concentrations, might be good candidates to increase photoprotection. The choice seems random.
We chose DBS and HSA, efficient and well-known gelators, largely used in cosmetic formulations. These organogelators have also been the subject of our previous research studies. Their UV protection abilities were observed consequently.
- While the title and purpose affirm emphasis on photoprotection, the majority of the work is on mechanical properties
We have taken a careful note of your suggestion, and the title and purpose were modified.
- Only 3 formulations were tested here, and the one that showed photoprotective properties was tested only at one concentration
Samples were selected according to a selecting formulation study based on the sensory panel survey results. Then, to the final four selected lipsticks, the in vitro SPF values were estimated. In future studies, we envisage developing particularly the photoprotecting properties of LMOG samples varying the proportions and the nature of their ingredients.
- No discussion as to what makes a gelling material photoprotective is included
Please find in Effect of organogelators on in vitro SPF and UVA-PF:
L2 formulation containing 10 % 12-HSA as organogelator showed higher UV-absorbing ability in comparison with the other formulations containing DBS as organogelator (L1 and L3) or no organogelator (L4, L5). This difference in absorbance intensity could be related to the optical properties of the sticks, and in particular to their diffusing capacity. Indeed, organogels contained 10 wt % of 12-HSA, which made them very diffusing (transfused aspect) and increased their capacity to absorb the UV radiation.
- The statement in the conclusions section "design photoprotective lipsticks" is inaccurate since no design was involved here.
The suggestion was taken into account.
Reviewer 2 Report
The manuscript describes an interesting study on the use of LMOGs as structuring agents in lipsticks. Oleogelators are more commonly used in cosmetic applications, so this is an interesting application with strong commercial potential. Overall, the study was interesting and well designed to evaluate the effect of two different LMOG molecules as partial substitutes for waxes. I view this work as a useful contribution to the field; however, there are some issues with the manuscript and interpretation of experimental data that need to be addressed prior to acceptance. If these points (outlined below) can be addressed, I feel this study would be suitable for publication in the special issue Gels Horizons. Comments are provided in the accompanying Word document.

Author Response
Manuscript ID Gels-1255378
Dear Reviewer,
Thank you for giving us the opportunity to submit a revised draft of the manuscript "Preliminary Study of the Potential Photoprotective Effect of Organogel-Based Lipstick Formulations: Texture Analysis, Rheological, Thermal and Sensory properties" for publication in Gels. We appreciate the time and effort that you and the reviewers dedicated to providing feedback on our manuscript and are grateful for the insightful comments on and valuable improvements to our paper. We have incorporated most of the suggestions made by the reviewers. Those changes are written in red within the manuscript. Please see below, in red, for a point-by-point response to the reviewers’ comments and concerns.
Response to Reviewer 2 :
General comments:
-The authors pitch this study as an investigation into the photoprotective effects of lipsticks containing LMOGs. However, the photoprotective aspect ends up being a relatively small component of the overall study. If additional experiments had been conducted to evaluate how to optimize or improve the photoprotective properties, this angle might be more appropriate. This is not an issue per-se, but it might be more appropriate to revise the title and introduction to reflect the overall study, as opposed to highlighting this one aspect that is only addressed with a single experiment for a select set of formulations, which in the end do not provide sufficient protection to meet minimum criteria to be classified as having photoprotective properties. Further, the only formulation which approached this minimum level was the formulation containing 10% LMOG (12-HSA), which seems quite high, considering 12-HSA can form a 3-dimensional structure at ~1 wt% structurant.
The title and the introduction were revised according to your recommendation. Concerning the 12-HSA wt %, the sample was selected according to a selecting formulation study based on the sensory panel survey results. Decreasing the 12-HSA wt % leads to disturbing the expected rigidity and spreadability of the resulting lipstick. In our future studies, we will develop the photoprotecting properties of LMOG samples varying the proportions and the nature of their ingredients.
-Is it common for lipsticks to contain castor oil? As castor oil is rich in ricinoleic acid, the hydroxyl groups may interfere with the organogelation ability of 12-HSA, as it has been shown that polar molecules in the solvent can impede formation of the SAFiN network. Perhaps a much lower 12-HSA content could be used if castor oil could be replaced with another oil source (?).
In lipstick formulations, castor oil is usually used for its softening and anti-chapping properties. In accordance to your suggestion, in our future research works, we may consider samples contained other oils or substances poor in polar molecules, while maintaining the required demulcent properties.
-It may also be useful to note the main structuring agents of commercial lipsticks, and how these contribute to functionality in the introduction. These are important aspects to consider when incorporating various oleogelators.
The main structuring agents of commercial lipsticks are mentioned in the introduction and their component functional categories, in the table 1 :
Lipsticks are commonly composed of a lot different ingredients, such as vegetable oils (castor oil, almond oil), mineral derivatives (vaseline oil, white petrolatum), pigments and waxes which are not only for aesthetic purposes but also can act as bioactive agents for environment extreme weather or UV protection.
-The statistical analysis provided in the graphs is useful, but sometimes difficult to quickly interpret (such as when referring back to the Figure). Could the authors also incorporate letter differences to indicate al significant differences in addition to the current representation which indicates the extent of significance between various formulations?
Your suggestion has been taken into consideration. In figure 2, different lower-case letters indicate a significant difference for hardness, firmness or stiffness between days of the same formulation at 25 °C and different capital letters indicate a significant difference between different days of the same formulation at 45 °C (p<0.05). In figures 1 and 3, bars that do not share similar letters denote statistical significance, p<0.05.
General improvement of grammar and English needs improvement (predominantly in the Introduction). Some examples:
Line 51 and elsewhere: The term ‘pay-off’ seems to be a term used in the cosmetic industry, but this was initially unclear to me. Can the authors please briefly describe this term (e.g., the amount of material transferred during application?). Additionally, please ensure consistency throughout document, as there are instances of ‘pay-off’ and ‘pay off’.
Please consider the term pay-off. The pay-off is defined in the material and method part : Pay-off is often used as an indirect measure of how well a product works from a consumer’s perspective.
Line 56: Should add “based ARE affected”
The sentence was corrected.
Line 57: Natural?
The right word is natural.
Line 60: ricinoleic ACID
"Acid" was added.
Lines 83-86: Sentence is awkward, and needs revision.
The sentence was revised as follows : Authors noted that the gelator structure influenced the gel microstructural organization through weak intermolecular interactions, which on large-scale, could affect its rheological properties (viscoelasticity and thixotropy).
Line 119: My be more clear to state “…was heated to either 125C, or 200C when formulated with DSB” (or something similar)
The sentence was revised as follow : The phase A was heated either 125 °C or 200 °C when formulated with DBS.
Specific comments:
Line 79: What is meant by stability in this context? Physical (no oil loss?), oxidative, consistency of mechanical performance, etc.?
Various studies have characterized the rheological, mechanical, physical stability and thermal behavior of these LMOGs from diverse formulations.
Line 144: What is “the PPD action spectrum”? Please clarify.
Pλ is the Persistent Pigment Darkening (PPD) action spectrum.
Line 152: Please report the value of the load cell (e.g., 5kg, 30kg). Also, how were the samples secured during the measurement? Particularly, if not secured, could this affect the needle probe test?
The value of the load cell is 5 kg. The samples were placed on the plates with a hollow molding allowing them to be positioned in a vertical position.
Line 203: At what specific strain amplitude within the LVR were the tests performed?
The stain amplitude of 1 % was applied.
Line 225: Spelling error: “Tuckey’s test” should be “Tukey test”
The spelling error was corrected.
Fig. 1B: Is L1 not considered in the statistical analysis? It does not appear to be compared to L2, which appears to have a highly significant increase in SPF.
Statistical analysis has been corrected accordingly.
Line 272-273: This statement is very broad, and it is difficult to tell if Fig. 2 supports this claim. For example, was there a significant effect of storage time on Hardness for L2 at 25 degC? Is the statistical significance from storage time displayed in these figures? Is the statistical test simply between temperatures of the same sample at the same storage time?
I believe the figure needs to be improved to better demonstrate the statistical analysis, or appropriate tables should be provided with this information.
Can the authors elaborate on why mechanical tests of the commercial control were not evaluated at the various time points?
The sentence (lines 272-273) was modified as follows: “The effect of lipstick ingredients, storage temperature and storage time during TPA were evaluated.” Statistical analysis was extended to include the effect of storage time on each formulation and for each mechanical tests (hardness, firmness, stiffness). Mechanical tests of the commercial control were not taken into account because the date of manufacture was not known since they were not manufactured in the laboratory. The effect of time in this case cannot be taken into account because of the bias it could induce.
Line 286: Please define “gf” (assumed to be grams of force?) Also please define RT at Line 289, or alternatively continue to explicitly state 25 degC.
"gf" is a gram-force unit. RT is replace by 25 °C.
Line 294: Please elaborate on what the structuring agent(s) was(were) in the noted reference.
Please consider the revised sentence: These results were previously confirmed in different studies reported by Esposito et al., wherein 12-HSA-based organogels heated up above Tgel temperature undergo a molecular reorganization due to disruption of physical interactions between organogelator molecules.
Line 297: The term ‘resilience’ for textural analysis generally considers the work during decompression relative to the work during compression. Therefore, stiffness does not seem to be an appropriate analog for resilience.
Please consider the revised sentence : Stiffness of the sample refers to its ability to resist elastic deformation during the bending action and is dependent on the elastic modulus (Young's modulus). At 25 °C and 45 °C, L4 was the less stiffer lipstick throughout the study (Figure 2B).
Lines 309-310: The similarity of L4 to control does not provide an appropriate comparison with L1-L3. Direct statistical analysis is required to make this claim. Please revise.
Lines 309-310 have been modified accordingly as follows: “L4 values at 25 and 45°C were significantly lower than L1-L3 lipsticks (P<0.0001), suggesting L4 and even L5 (having the lowest firmness values at day1) were the softest lipsticks during the study.”
Lines 311-313: Is the statistical analysis across storage time provided? As noted above, no indication of statistical comparison across time points seems to be present in Fig. 2. Additionally, the comment “as well as the control” seems confusing. Was the commercial control analyzed at various time points? This is not shown.
Statistical analysis was extended to include the effect of storage time on each formulation and for each mechanical tests (hardness, firmness, stiffness). Please consider the revised sentence : “At 45 °C, L1 and L3 formulations exhibited significantly softer texture throughout the weeks (p < 0.05).”
Line 314: The authors have not yet introduced the rheological data. Please do not discuss prior to introducing this section, or make an explicit note to the appropriate section.
Please consider the revised sentence : As expected, the presence and higher concentration of LMOGs enhance firmness. However, as temperature increased, the firmness of LMOG-based formulations with low organogelator concentration decreased, probably due to oil losses from the network [37].
Line 325: The term “long-lasting” does not match the sentence structure; perhaps “durability”?
Please consider the revised sentence : Although a lower pay-off results in low amount of product attributes (color, durability, coverage) transferred, a balanced low/high pay-off needs to be considered to reduce product consumption and undesired sensation (waxy buildup).
Line 330: What was the minimum requirement? Please state explicitly.
Regarding requirements, the lipstick should have line 325 at least a pay-off of 0.0001gm/cm2.
Line 333-334: The explanation/justification provided for difference in pay-off seems unclear. Are the authors stating the LMOG-based gels have a rougher surface? Do the authors have any measure of the difference in frictional force to justify the various significant differences between samples, or microstructural evidence to support this claim? As noted, commercial samples also contain waxes, which form platelet structures that could be envisioned to be rough. Likely the temperature effect also plays a strong roll. If the authors would like to discuss friction, it would be useful to report a measured value. A friction rig was used to quantify pay-off, so it would likely be straight-forward to determine the friction coefficient (possibly from the experimental data already collected).
It also seems curious that sample L4 was quite soft, but also had a low pay-off. Can the authors comment on this? Was the material soft, but elastic?
The suggestion was taken in account. To discuss about the difference in pay-off values, we need to realize complementary studies which could be the subject of further experiments.
Please consider the revised sentence: The higher pay-off values between LMOG-based and "classical" lipstick formulations could be explained by an higher frictional force change attributed to gels samples.
Line 343 and Fig. 4 caption: “LVE” should read “LVR”
The spelling error was corrected.
Line 343-356: The interpretation of the rheological behavior is unclear, and should be revised. I fail to see the initial increase in G’ or the two plateau regions noted by the authors. The description “a decrease in G’ in the region 45-140 degC does not accurately describe the data, where each sample seems to have a nearly constant G’ up to the melting point, upon which there is an abrupt melting event.
The interpretation of the rheological behavior was revised as follows :
The temperature sweeps are displayed in Figure 4 and were performed in LVR regime to assess the effect of organogels-based lipsticks between 15 °C and 150 °C on the dynamic elastic behavior (G′) and the dynamic viscous behavior (G″). The organogel-based lipsticks exhibited thermoreversibility and viscoelastic properties. According to the classical viscoelasticity theory, the initial rise in G′ is due to 3D network formation while onset sol-gel phase transition (Tgel) of each lipstick can be estimated from the abrupt increase in G′. Indeed, Tgel is determined by the G′/ G″ crossover point from G′ < G″ to G′ > G″ of each cooling curve. A crossover point of G′ and G″ is identified as the gel melting point (Tm). Upon cooling, the resulting disordered fluid-like state of the gel undergoes a sol-gel transition generating a viscoelastic solid-like structure of crystalline organogelator molecules [18]. L1 and L3 showed a crossover point at high temperatures underlying a more stable network with highest melting stability [12,45], while L2 and especially L4 and L5 were more heat amenable and prone to soften at low temperature.
It is also a bit surprising that no decrease in G’ is observed in any softening is observed in L4 or L5, where there were distinct differences in the large deformation behavior between measurements at 25 and 45 degC. Can the authors comment on this?
According to the results shown on figure 4, the L4 and L5 presented the gel-sol and sol-gel transition as those observed with the LMOG-based samples. We suppose that the setting of the temperature ramp (swift heating gradient) could interfere in the softening state of the samples shifting the G’ decreased values after 45 °C.
Fig 4 caption: Some symbols did not present correctly in caption (L1, L2, L3).
The symbols were corrected.
Table 2: Please add statistical significance, at least within columns, as this is discussed in the text (Line 365-366).
Statistical analysis was included for Tgel and Tm (rheology). Lines 365-366 was reformulated as follows: “It was also noted a small but significant increase in the Tgel value (p value < 0.05) and in a maximal G′ of 12.9 times, with the addition of DBS organogelator at 1.5 % (w/w) in the lipstick network (L3), compared with the control (L4).”
Line 368: Are the authors referring to the results in Fig 2A vs Fig 4/Table 2? If so, please state explicitly. Further, the results do not seem to directly correlate, as L2 was much firmer than L1 and L2 by large deformation, but weaker by small deformation. Can the authors also comment on this point if they would like to discuss the correlation between these two regimes, as different mechanisms are responsible for the mechanical response probed at these different length scales.
Line 368 refers to table 2, as mentioned above (line 364) and this has been added to line 368, as requested. Regarding lines 369-371, please consider the revised sentence : Higher G′ values are typically associated with stronger three-dimensional gelled network [42]. Mechanical and rheological properties were evaluated all along complementary experiments, each one is a separate quality to consider.
Line 373-377: It seems the authors have already discussed the crossover points in the previous paragraph (when G’ becomes lower than G’’). Therefore, this section seems redundant. Please revise.
The crossover points were already discussed previously, but it is important to comment the figure S3 (see complementary section) introducing the tan δ parameter.
Line 385-387: The authors have stated that the products were acceptable by consumers, but this statement is made before providing experimental data. Please revise.
The suggestion was taken in account : Nevertheless, all samples showed acceptable thermal stability and could be considered as heat-resistant lipsticks.
Line 398: The ‘consensus’ amongst products was not noted in the methodology. Did panelists evaluate the products in a group setting, or individually? I.e., was agreement determined based on individual scoring, or could panelists have been influenced by a group setting?
Please consider the revised sentence : The performance of the panel was evaluated through different criteria: consensus (individual product evaluation) and discrimination between product and attributes.
In Line 396/397, ‘variable’ should be replaced with ‘value.
The spelling error was corrected.
Line 452: The overall sensory analysis is quite interesting. Can the authors complete this section with a broad take-away for the reader?
This suggestion has been taken into account. Lines 542-544 : “Overall, the main discriminative attributes for the analyzed lipsticks were : greasiness and glossiness. The experimental and sensory hardness parameters were in total agreement for the hardest and softest formulation.”
Line 466-467: As noted above, the authors POSTULATE that friction causes differences in pay-off. This should be revised, or supporting data should be provided.
The suggestion was taken in account.
Final point of conclusion states that LMOGs can be used to improve photoprotective capacity of lipsticks; however it was demonstrated that this is highly gelator-specific. Perhaps this statement should be revised to note that a balance must be struck between structuring capacity, functionality, and photoprotective effects.
The sentense was revised : By combining sensory attributes (glossy) and instrumental parameters (hardness, firmness), we were able to cluster the formulations. Considering the overall mechanical, thermal and photoprotection attributes, LMOGs-based formulations have a great potential to replace part of waxes in lipsticks, while offering tunable textural parameters to target a large number of customers.
